# Evaluation of Robotic Systems on Cytotoxic Drug Preparation: A Systematic Review and Meta-Analysis

**DOI:** 10.3390/medicina59030431

**Published:** 2023-02-22

**Authors:** Sangyoon Shin, Jamin Koo, Suk Won Kim, Seungyeon Kim, So Yeon Hong, Euni Lee

**Affiliations:** 1College of Pharmacy & Research Institute of Pharmaceutical Sciences, Seoul National University, Seoul 08826, Republic of Korea; 2College of Pharmacy, Dankook University, Cheonan 31116, Republic of Korea; 3Department of Pharmacy, Seoul National University Bundang Hospital 82, Gumi-ro 173, Bundang-gu, Seongnam-si 13620, Gyeonggi-do, Republic of Korea

**Keywords:** hospital pharmacy, robotic system, safety, cytotoxic drugs

## Abstract

*Background and Objectives*: With the increased prevalence of patients with cancer, the demand for preparing cytotoxic drugs was increased by health-system pharmacists. To reduce the workload and contamination of work areas in pharmacies, compounding robots preparing cytotoxic drugs have been introduced, and the use of the robots has been expanded in recent years. As reports on the comprehensive and quantitative evaluation of compounding robots remain lacking, a systematic review and meta-analysis were conducted to provide descriptive and quantitative evaluations of the accuracy of preparing injectable cytotoxic drugs. *Materials and Methods*: A systematic review and meta-analysis were conducted using published studies up to 2020. To identify eligible studies, PubMed, EMBASE, and Cochrane Library were used. All studies reporting the outcomes relevant to drug-compounding robots such as accuracy, safety, and drug contamination were included. Outcomes from included studies were descriptively summarized. Drug contamination by the robot was quantitatively analyzed using the odds ratio (OR) with a 95% confidence interval (CI). The risk of bias was assessed using the Risk of Bias Assessment tool for Non-randomized Studies (RoBANS). *Results*: A total of 14 compounding robot studies were eligible for review and 4 studies were included in the meta-analysis. Robotic compounding showed failure rates of 0.9–16.75%, while the accuracy range was set at 5%. Two studies reported that robotic compounding needed more time than manual compounding, two reported that robotic compounding needed less time, and one just reported preparation time without a control group. In a meta-analysis regarding the contamination of the compounding area, manual compounding was associated with lower contamination, although the result was not statistically significant (OR 4.251, 95% CI 0.439–51.772). For the contamination of infusion bags, the robot was associated with lower contamination (OR 0.176, 95% CI 0.084–0.365). *Conclusions*: Robotic compounding showed better accuracy than manual compounding and, without control groups, showed a high accuracy rate and also reduced the risk of drug contamination and compounding workload. The preparation time of the robot was not consistent because the type of robot and introduced system were different. In conclusion, robotic compounding showed mixed results compared to the manual compounding of drugs, so the system should be introduced considering the risks and benefits of robots.

## 1. Introduction

According to the World Health Organization, cancer is the second leading cause of death worldwide [1]. Advancements in medical technology and public health have enabled early detection of cancers and subsequent treatment with chemotherapy. In addition, increasing demand for chemotherapeutic agents has been reported [2] and the projected drug market for anti-cancer agents is expected to reach USD 135.5 billion by 2025 in terms of global sales [3].

Growing demands for anti-cancer drugs inevitably would necessitate an increase in pharmacists’ workload for preparing injectable cytotoxic agents. In addition to the increased pharmacists’ workload and occupational stress related to chemotherapy preparation [4], potential safety hazards associated with the handling of cytotoxic agents were also considered as established causes for concern [5]. The new USP chapter 800 outlines standards to protect against exposure to cytotoxic drugs; therefore, concerns about potential harm should be resolved [6]. Despite strict health and safety control of the working environment and drug preparation procedures, the contamination of work areas with anti-cancer drug residues was constantly reported [7].

An increased expectation of continuous quality improvement to reduce dispensing errors, especially regarding patient-specific dosages of chemotherapeutic agents, has led to robotic compounding as a new solution [8]. Therefore, with the objective to mitigate potential safety risks associated with, and improve the accuracy and safety of, anti-cancer drug compounding, robots for injectable preparations have been used in many countries in Europe, North America, and Asia, including South Korea [9,10].

A drug-compounding robot uses arms to prepare injectable drugs automatically in an aseptic field. When a prescription is entered, the chemotherapeutic agent to be dispensed is recognized and confirmed, thus ensuing automated drug compounding according to the manufacturer’s algorithm. The robot is equipped with a scale that checks the accuracy of the weight of a prepared product considering the specific gravity of the drug and the reconstitution fluid [2]. Following the commercialization of the first one-arm compounding robot called CytoCare^®^ in Europe, a series of other compounding robots, including I.V. Station Onco^®^, APOTECAchemo [11,12], and KIRO^®^ Oncology, a two-arm robot [13], have been introduced to prepare compounding cytotoxic agents. Although a published systematic review by Batson and colleagues provided narrative summaries on the accuracy, efficacy, contamination, and cost savings of commercially available compounding robots [9], there remains a lack of quantitative evaluations and up-to-date reviews using the latest studies. Therefore, we performed a systematic review and meta-analysis to examine the performance of compounding robots in terms of their accuracy in compounding chemotherapeutic injectable drugs and the safety of pharmacy workstations.

## 2. Materials and Methods

### 2.1. Search Strategy and Data Sources

We conducted a systematic review and meta-analysis according to the Preferred Reporting Items for Systematic Reviews and Meta-Analyses (PRISMA-P) guideline [14], using the following databases: PubMed, EMBASE, and the Cochrane Central Register of Controlled Trials. All eligible studies were identified up to July 2020, and bibliographies from retrieved articles were scrutinized and manually searched for further relevant studies. To search the published literature on compounding robots, the following keywords were used in combinations: ‘compounding’, ‘robotics’, and ‘automation’ (Appendix A).

### 2.2. Eligibility Criteria and Study Selection

The selection of relevant articles was conducted using eligibility criteria based on the patient, intervention, comparison, outcome, and study design (PICO-SD) framework. Studies related to pharmacists or pharmacy technicians who operated compounding robots or to patients who received cytotoxic agents prepared using compounding robots were eligible for inclusion. Comparative studies evaluating compounding robots and other compounding methods, as well as descriptive reviews of compounding robots, were also eligible. Only articles published in English were included in the final selection. Studies related to robotic surgery, non-robotic compounding, and computerization of workflow for quality improvement were excluded. Non-original research articles, review articles, and articles not available as full text were also excluded (Figure 1). Study designs eligible for selection included randomized controlled trials, cohort studies, cross-sectional studies, case-control studies, and quasi-experimental studies.

Outcomes related to the performance of the compounding robot were measured in terms of accuracy, drug preparation time, and safety, including drug contamination of infusion bags or gloves worn by the robot operator. Of the outcomes, only contamination after compounding and contamination of infusion bags were appropriate for quantitative analyses.

One researcher (J.K.) identified and retrieved articles following the search strategy. A second researcher (S.W.K.) then assessed the relevance of the retrieved articles, and a third researcher (S.S.) verified the whole process. After removing duplicate articles, two researchers (J.K. and S.S.) independently selected studies by reviewing titles, abstracts, and full text according to the eligibility criteria. Any disagreement on study selection was resolved with a discussion among the researchers and a third researcher (E.L.). Based on our search results from the systematic review of compounding robots, articles that met the inclusion criteria for the meta-analysis were identified for further analyses.

### 2.3. Data Extraction and Quality Assessment

Extracted data included bibliographical items and study characteristics such as year of publication, country of study, funding acquisition, study design, study period, name of the compounding robot, and robot operator status (e.g., pharmacist or technician). We extracted data related to the accuracy, contamination of the compounding area and infusion bags, and time required for preparation as the main outcomes. Other data for the secondary outcomes were also extracted, including cost, drug exposure, ‘failure mode, effect, and criticality analysis’ (FMECA), and several failure factors.

To assess the quality of the included studies, the Risk of Bias Assessment tool for Non-randomized Studies (RoBANS) tool was used [15]. The tool is usually used to assess the risk of bias in non-randomized studies such as observational studies. The RoBANS tool assesses the risk of study bias in six domains: selection of participants, confounding variables, measurement of intervention, blinding of outcome assessment, incomplete outcome data, and selective reporting [15]. Based on each RoBANS domain, all included studies were evaluated as having a ‘low’, ‘high’, or ‘unclear’ risk of bias. Two researchers (J.K./S.K.) evaluated the quality of the studies, and any discrepancies were resolved with a consensus including all authors.

### 2.4. Data Analysis

In this study, the accuracy of a robotic system was defined as the system failure rate of compounding cytotoxic agents by the robot. With a 4% [16], 5% [2,17,18,19,20,21], or 10% [17] accuracy threshold evaluated by the studies, the accuracy of robots was summarized as the number of failures or the proportion of failures from each study. Drug contamination was defined as the rate of detecting cytotoxic agent residues; the differences in the concentration of cytotoxic agents before- and after-operating robots were measured on the compounding area, operators’ gloves, or infusion bags. The odds of contamination using a robotic system were compared to the odds using manual compounding. Microbial contamination was defined as the rate of detecting colonies using a media fill test generated by the total number of compounding procedures. Statistical heterogeneity across studies was assessed using the *I*^2^ statistic, with an *I*^2^ value of over 50% indicating heterogeneity. Data analyses were performed using Comprehensive Meta-Analysis version 2 (Biostat, Englewood, NJ, USA).

## 3. Results

### 3.1. Selection of Studies

A total of 927 studies were initially identified using the databases Medline, Embase, and the Cochrane Library. Two additional studies were retrieved from a pre-existing systematic review paper [9] (Figure 1). Of these 929 studies, 71 duplicates were removed. By evaluating titles and abstracts, we identified 65 studies and excluded 793 non-relevant publications which described robotic surgery, automated formulation, automated injection formulation, automated workflow, and studies in non-pharmacy settings. Of the 65 studies, 14 were selected for systematic review, of which 4 were eligible for the meta-analysis (Figure 1).

### 3.2. Outcomes

Study characteristics, as well as main and secondary outcomes, of the 14 studies included in our systematic review are summarized in Table 1.

#### 3.2.1. Accuracy of Compounding Robots

The accuracy of the robotic system was measured in seven studies [2,16,17,18,20,25,27]. Of these, two studies [2,20] compared the accuracy of manual drug compounding versus that of robotic compounding and found that the latter had higher accuracy. Seger et al. [2] reported failure rates of 12.5% with manual compounding, compared to only 0.9% with robotic compounding. By contrast, the remaining five studies [16,18,20,25,27] measured only robotic system accuracy, with no control group. Failure rates of 2.1–16.75% [17,18,25,27] and 1.15–4.10% [16,17,27] were reported when the dose accuracy range was set at 5% and increased to 10%, respectively (Table 1).

#### 3.2.2. Meta-Analysis on Contamination

Contamination by robotic compounding was categorized as drug contamination in compounding areas and on infusion bags and microbial contamination. Five studies assessed drug contamination: Iwamoto et al. [20], Kramer et al. [24], Schierl et al. [21], Sessink et al. [22], and Buning et al. [23]. Three studies assessed microbial contamination: Geersing et al. [26], Jobard et al. [27], and Sabatini et al. [28].

#### 3.2.3. Drug Contamination

Drug contamination was assessed in compounding areas, on gloves worn by robot operators, and on infusion bags. In studies including a control (i.e., manual compounding), drug contamination was also assessed on work surfaces used for manual compounding, gloves worn for manual compounding, and infusion bags handled manually during compounding. Although four studies (Sessink et al. [22], Iwamoto et al. [20], Schierl et al. [21], Buning et al. [23]) reported outcomes regarding contamination of the compounding area, the Sessink et al. study was excluded because of accidental drug spillage [22].

A meta-analysis of drug contamination showed that manual compounding was associated with lower contamination rates in compounding areas compared to robotic compounding, although this result was not statistically significant (*p*-value = 0.257) (Figure 2). Using the random effects model, we obtained an *I*^2^ value of 57.583.

Four studies using the robotic system APOTECAchemo for compounding were included in the meta-analysis regarding the contamination of infusion bags [20,21,23,24]. Two studies investigating the robotic system CytoCare^®^ were excluded due to accidental drug spillage and excessively large standard deviations [22,24]. We performed the meta-analysis using the fixed model, with an *I*^2^ value of 48.582. Results showed that robotic compounding was associated with significantly lower contamination rates compared to manual compounding (*p*-value <0.001) (Figure 3).

#### 3.2.4. Microbial Contamination

In the three studies [26,27,28] that assessed microbial contamination, the surface contamination level in robotic compounding was measured. In addition, Jobard et al. [27] measured contamination of the final product as well as operators’ gloves worn during drug compounding, and Sabatini et al. [28] measured contamination of the final product when robotic compounding was completed. Wipe and swab samples were incubated in suitable culture media, and the number of colonies were evaluated (Table 2). Some microbial contamination was found in Geersing et al.’s study [26]. In addition to wipe sampling, contamination of the environment was also measured in key relevant locations using active air sampling and particle counting. The assessment results showed that the compounding environment met the aseptic quality standard of European GMP class A or ISO class 5 (Table 1).

#### 3.2.5. Drug Preparation Time

Four studies [2,17,18,19] included drug preparation time as a study outcome (summarized in Table 1). Drug preparation time was obtained by measuring either medication turnaround time (TAT) or medication preparation time (PT). TAT is the time taken from a pharmacy technician receiving a drug order and starting the compounding to a pharmacist completing their inspection. PT is the time taken for a pharmacist or pharmacy technician to prepare a drug. Bhakta et al.’s study [18] showed that a significant mean decrease of 10 min in TAT was observed with robotic compounding, compared to manual compounding. However, there was no significant difference in PT between the two compounding modes. On the other hand, Seger et al. [2] found longer PT and TAT with robotic compounding than with manual compounding. In the study by Heloury et al. [19], there was no significant difference in PT between robotic and manual compounding of patient-specific drugs; however, in the case of bulk production, a significant decrease in PT was observed with robotic compounding. In the study by Nurgat et al. [17], PT was based on robotic and manual throughput in terms of the number of doses produced. The authors showed that the same throughput was obtained with 2–3 h of manual compounding compared to 7 h of robotic compounding.

### 3.3. Secondary Outcomes

Secondary outcomes assessed using the studies include costs, drug exposure, the sum of failure mode, effects, and criticality analysis (FMECA), and several failure factors where the dispensing failure occurred (Appendix A). Costs comprised the full-time equivalent (FTE) of manpower, ancillary costs, and drug savings. From studies by Bhakta et al. [18] and Seger et al. [2], it was possible to save more than USD 100,000 (USD 129,477 and USD 115,500, respectively) in drug or ancillary costs per year. FTE, a value that compares how many pharmacists or technicians can be replaced with automated technology, was measured by Chen et al. [25] and Nurgat et al. [17]. In the study by Chen et al. [25], robot use reduced the workload of pharmacists by 2 h, but not the FTE because the personnel’s time spent operating the robot had to be included. In the study by Nurgat et al. [17], it was noted that 7 h of robotic work was equivalent to 2–3 h of manual compounding.

The degree of drug exposure can be obtained using the FMECA method. We found three studies that included the FMECA approach [2,19,25]. According to Chen et al. [25], robot use was not associated with a reduced risk of drug exposure compared to manual compounding. Heloury et al. [19] showed that the FMECA score for drug exposure with robot compounding was 404 points lower than that with manual compounding. In Seger et al.’s study [2], 82 safety events were recorded out of 1421 preparations during manual compounding, compared to 35 out of 972 preparations during robotic compounding. In their study, Yaniv et al. [16] reported cases of dispensing failure due to mechanical problems (most frequent), human errors, and interface issues.

### 3.4. Risk of Bias

The quality of the included studies was assessed, as shown in Figure 4 and Appendix A. Missing outcome data were not detected in most of the studies, and because the reason for any missing data was given by the study authors, studies with incomplete outcome data were considered as having a low risk of bias. With respect to confounding variables, all studies were considered as having a low risk of bias as they contained only a few confounding variables. On the other hand, the majority of studies were considered as having a high risk of bias in terms of blinding of outcome assessment because data extraction was conducted by the researchers.

This section may be divided into subheadings. It should provide a concise and precise description of the experimental results, their interpretation, as well as the experimental conclusions that can be drawn.

## 4. Discussion

One of the strengths of our study was the methodologic approach of not only conducting a comprehensive systematic review but also focusing on performing a meta-analysis to provide quantitative measures using full-text original articles published on robotic performance in drug compounding, as well as on outcome measures related to the safety of healthcare staff operating robotic systems for compounding cytotoxic agents. Of the 14 studies from our SR, 8 studies evaluated accuracy in various outcome measures including TAT, preparation time, and contamination. Although Batson and colleagues provided fairly comprehensive reviews on the performance of compounding robotic systems [9], the study provided a narrative summary without providing collective and comparable indices. As our study included recently published studies introducing new brands of robots, we believe the findings from our study could add value to the literature by providing a more comprehensive summary of the current literature and quantitative measures that can serve as points of reference.

Based on two studies [2,20], we found that the accuracy of robotic compounding was significantly higher than that of manual compounding (Table 1). In addition, the *United States Pharmacopeia* recommends the acceptable range of most compounded preparations as ±10%. [17,29] However, given that the failure rates of compounding robots were reported to be around 2% with a strict dose accuracy range set at 5%, robotic compounding can be considered reliable in terms of accuracy.

There are reports of occupational exposure of healthcare workers to anti-cancer drugs resulting in a higher incidence of cancer, compared to other occupations not carrying such exposure risk [7]. Findings from our meta-analysis on drug contamination indicated that the risk of direct exposure of healthcare workers to anti-cancer drugs through drug contamination from the compounding process could be reduced by the use of compounding robots. In the case of the manual handling of infusion bags, drug residues are likely to be found on the surface of the final product, such that there will still remain a risk of direct exposure to anti-cancer drugs despite the worker wearing protective equipment. For example, the risk of vial surface contamination would be higher with a drug such as cyclophosphamide that is known to undergo a high level of vaporization [30]. Taken together, therefore, although the working environment may be similar, reduced contamination of infusion bags with the use of automated compounding suggests the superiority of robotic compounding to manual compounding for the occupational safety of operating personnel. Although a protective mat absorbing any spillage of anti-cancer drugs during manual compounding can be beneficial in reducing the risk of contamination, findings from our meta-analysis indicate that drug contamination of the compounding environment was not related to contamination of the final product, i.e., the infusion bags, which, could, in turn, surmise that automated compounding has a better safety profile.

In our systematic review, we also assessed PT. In most cases, manual compounding was associated with a more efficient and faster PT than robotic compounding, which is similar to the review by Batson et al. [9]. However, a recently published study by Bhakta et al. reported no difference in PT between manual compounding and robotic compounding and decreased TAT with the robotic system [18]. We believe that the reduced TAT indicated a decrease in preparation time when pharmacists used automated technology, and the results were corroborated with a study by Seger et al. [2]. Taken together, the results of our meta-analysis suggest that a robotic system could contribute to reducing pharmacists’ workload in the compounding of cytotoxic agents while enabling a safer working environment for pharmacists and other relevant staff. In addition, we believe that automated technology would be useful in ‘freeing’ time for pharmacists to focus on providing cognitive services for improved patient-centered care.

The findings from Bhakata et al. and Seger et al. reporting cost savings of compounding robot [2,18] are in contrast to those from Chen et al. reporting no difference in FTE savings between manual compounding and compounding robot [25], which may indicate that the compounding robot is useful compared to manual compounding. However, our study could only provide descriptive and narrative results because few outcomes about cost and benefit were reported. Furthermore, important economic outcomes such as the set-up cost of the robot, the cost of the disposable equipment used in robotic compounding, and the time to dismantle and clean up were not available for our review. Thus, further studies presenting practical issues are needed to compare the costs and benefits of compounding robots in a real-world setting.

It is conceivable that the introduction of this new technology, namely the compounding robot, could generate a new type of error. If human error is the main source of errors in anti-cancer drug compounding, then the introduction of robotic compounding could also give rise to a variety of errors, including defects in the robot itself, mechanical errors related to the scale calibration of the robot, and errors related to the interface used in robotic compounding [16]. Therefore, it is essential to consider and understand the possible errors that are likely to arise from this new technology so appropriate measures can be taken to mitigate the risk of errors by, for example, proactively running educational events aimed at raising awareness of the various types of robotic compounding systems and their pros and cons, including potential sources of errors, among the relevant automated technology community, including pharmacists.

### Limitations

Our study has a few limitations that should be considered when interpreting its findings. Except for the study by Bhakta et al. [18], which explicitly described the study design, all other included studies did not provide specific details, which meant we had to rely on our best judgement to determine the study design as part of our inclusion criteria.

The accuracy of compounding robots was reported in the majority of studies, but a method to evaluate accuracy was not unified. In addition, every study suggests its own definition of the accurate dose of a drug and threshold limits, so that objective and comparative evaluation of accuracy between studies could not be performed, and thus findings were presented as a descriptive summary. We believe the possibility of the Hawthorne effect could not be ignored in both the control group and the experimental group.

With regard to the sampling location for measuring contamination, we attempted to combine all studies reporting the contamination source as the robot arm that directly injected the drug. Although some studies described various swab sampling sources, we believe, however, our findings reflect appropriately the overall contamination rate across the robotic systems, given a common contamination source—the robot arm. Similar to this limitation, the method to evaluate contamination was not standardized. Even if some compounding robots were introduced, the system to prepare drugs differs among hospitals. A different definition of contamination by a hospital should be added as an additional limitation when quantitatively synthesizing the results. The interpretation of our findings on contamination should be also limited as additional contamination sources were not evaluated. For example, contamination associated with cleaning up the robot before and after the compounding can be hazardous to pharmacists or technicians. Further studies about comprehensive safety outcomes in operating robotic system are needed. Variability in swab sampling frequencies could be a source of bias in the evaluation of contamination. Iwamoto et al. [20] reported two swab samplings for each evaluation of contamination in the compounding area after manual compounding, as compared to two other studies [21,23] which reported a collection of 12 samplings. In our study, we restricted the evaluation of contamination to APOTECAchemo. However, further studies are needed for more objective assessments of the contamination rates that would include a wider range of robotic systems.

There are numerous considerable factors when comparing costs and benefits between manual and robotic compounding of drugs. In this article, only three studies reported the cost of robotic compounding limited to costs associated with doses and workforce. However, external factors related to introducing the robots, such as time for standardization of the robot into the system within the hospital, cleaning up costs, and dismantling and assembling the robot were not included. Further studies should include economic outcomes that can overcome these limitations.

In spite of these limitations, we believe that our study findings could serve as a point of reference in terms of safety for healthcare workers, including pharmacists, who operate robotic systems for day-to-day compounding of cytotoxic agents, with consideration of the risks and benefits of using automated technology.

## 5. Conclusions

Robotic compounding was more accurate than manual compounding and reduced the risk of drug contamination on the compounding area, infusion bags, and operators’ gloves during compounding, which would contribute to improved safety compared to manual compounding. A comparative evaluation of the efficacy and cost savings of robotic compounding could not be made due to insufficient information in the literature.

The introduction of compounding robots could help pharmacists reduce their compounding-related workload so they can focus more on providing cognitive services. Before automated compounding can become widely established, pharmacists should be educated on the risks of potential errors that can arise with the use of compounding robots so they can adopt appropriate precautionary measures to mitigate those risks.

## Figures and Tables

**Figure 1 medicina-59-00431-f001:**
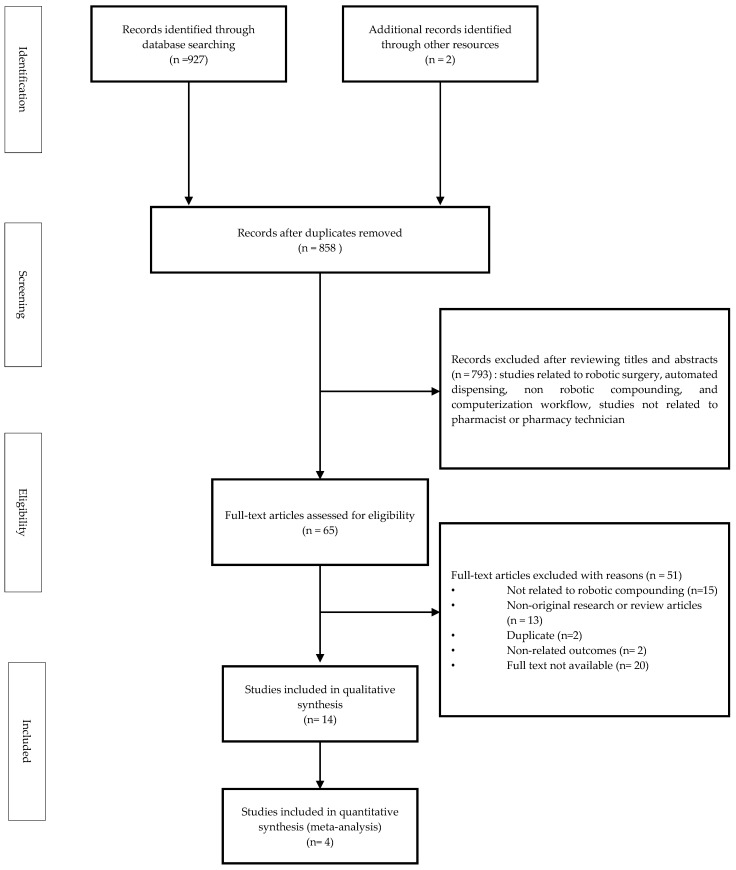
Flow chart showing study identification and selection.

**Figure 2 medicina-59-00431-f002:**
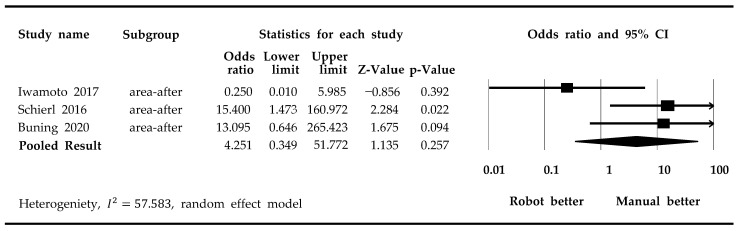
Forest plots of the rate of contamination on compounding area [20,21,23].

**Figure 3 medicina-59-00431-f003:**
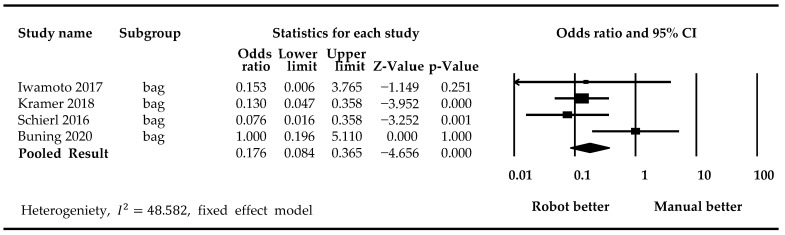
Forest plots showing the rate of contamination on infusion bags [20,21,23,26].

**Figure 4 medicina-59-00431-f004:**
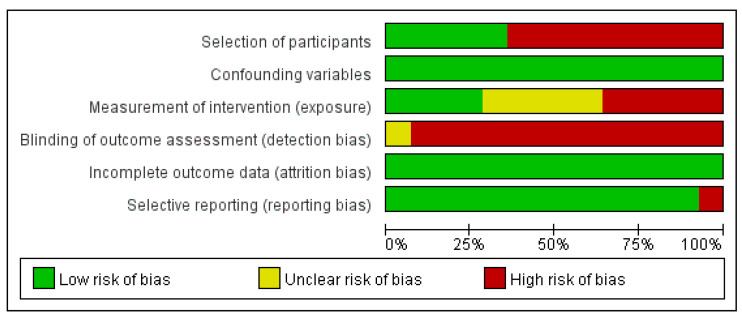
Risk of bias presented as percentages.

**Table 1 medicina-59-00431-t001:** Study characteristics and results of the systematic review.

Quasi-Experimental Study				
Study	Robot Brand	Outcomes	Results	
			Manual	Robot
Bhakata et al., USA, 2018 [18]	i.v.STATION	TAT ^†^ (mean, standard deviation)	64.1 ± 27.9 min	53.2 ± 2.2 min
		Accuracy	Not measured	11 failures out of 525 preparations
Heloury et al., France, 2019 [19]	KIRO	Preparation time	12 min 26 s ± 5 min 2 s per preparation	18 min 30 s ± 15 min 15 s per preparation
Iwamoto et al., Japan, 2017 [20]	APOTECAchemo	Accuracy (mean, coefficient variation)	FU ^¶^: 1.20%, 1.46% CPA ^¶¶^: 1.70%, 2.20%	FU: 0.83%, 1.04% CPA: 0.52%, 0.59% *
		Preparation time	Not measured	5.57 min per preparation for ready to use drug 6.11 min per preparation for lyophilized drug
		Contamination • Compounding area • Gloves • Infusion bag	1 out of 2 preparations 0 out of 3 preparations 2 out of 8 preparations	2 out of 10 preparations 0 out of 3 preparations 0 out of 8 preparations
Nurgat et al., Saudiarabia, 2015 [17]	CytoCare	Accuracy • Error range ± 5% • Error range ± 10%	Not measured	812 failures in 4846 doses 192 failures in 4846 doses
		Preparation-time (median, range)	IV syringe: 2.38 min (0.91–6.98 min) IV bag: 3.10 min (1.0–5.0 min) Glass bottle: 5.30 min (3.10–9.90 min)	Average number of doses prepared by the robot in a seven-hour shift equals doses that manually produced in a two to three-hour
Schierl et al., Italy, 2016 [21]	APOTECAchemo	Contamination • Surface • Gloves • Infusion bag	12 out of 21 samples 4 out of 7 samples 14 out of 20 samples	15 out of 21 samples 1 out of 8 samples 3 out of 20 samples
Seger et al., USA, 2012 [2]	CytoCare	Accuracy	23 failures out of 184 preparations	1 failure out of 110 preparations
		Preparation time (mean)	7 min 24 s per preparations	10 min 51 s * per preparation
Sessink et al., Belgium, 2015 [22]	CytoCare	Contamination • Air • Infusion bag • Inside of robot • Outside of robot	0 out of 7 samples Not measured Not measured Not measured	1 out of 5 samples 4 out of 8 bags 14 out of 23 samples 4 out of 20 samples
Buning et al., Netherland, 2020 [23]	APOTECAchemo	Contamination • Compounding area • Gloves • Infusion bag	0 out of 12 before-cleaning samples 1 out of 9 samples 3 out of 80 samples	5 out of 15 before-cleaning samples 1 out of 19 samples 3 out of 80 samples
Krämer et al., Germany, 2018 [24]	CytoCare and APOTECAchemo	Contamination • Infusion bag	23 out of 30 preparations	Cytocare: 29 out of 30 preparations APOTECAchemo: 18 out of 60 preparations
**Case-series study**				
**Study**	**Robot brand**	**Outcome**	**Result**	
Chen et al., Taiwan, 2013 [25]	CytoCare	Accuracy	123 failures out of 1028 preparations	
Geersing et al., Netherland, 2019 [26]	APOTECAchemo	Contamination ^‡^ • Particle count • Inside of robot • Surface of robot	Meets Class A limits Meets Class A limits Meets Class A limits	
Jobard et al., France, 2020 [27]	KIRO	Accuracy	4 failures out of total 150 bags 1 failure out of total 96 syringes 1 failure out of total 24 elastomeric pumps	
		Contamination • Air sampling • Surface • Gloves • Infusion bag	Satisfy ISO 5 2 out of 42 media fill tests 2 out of 24 media fill tests 0 out of 18 media fill tests	
Sabatini et al., Italy, 2018 [28]	APOTECAchemo	Contamination	0 out of 435 media fill tests	
Yaniv et al., USA, 2013 [16]	APOTECAchemo	Accuracy	85 dose issues ^§^ out of 7384 preparations	

^†^ TAT (turnaround time), ^¶^ FU (fluorouracil), ^¶¶^ CPA (cyclophosphamide), * *p* < 0.05, ^‡^ Contamination was assessed using the GMP annex 1 2008 criteria, ^§^ Issues which require manual addition of drug because of insufficient amount of drug.

**Table 2 medicina-59-00431-t002:** Microbial contamination presented as colony detection rates.

Study	Compounding Area (%)	Operators’ Gloves (%)	Infusion Bags (%)
Jobard et al. [27]	4.76	8.3	0
Sabatini et al. [28]	0	Not measured	0
Geersing et al. [26]	11.36	Not measured	Not measured

## Data Availability

No new data were created or analyzed in this study. Data sharing is not applicable to this article.

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
