# Peer review of "Evaluation of Robotic Systems on Cytotoxic Drug Preparation: A Systematic Review and Meta-Analysis"

_medicina, 2023, doi:10.3390/medicina59030431_

Round 1
Reviewer 1 Report
Thank you for the opportunity to review this interesting study. The information reported here is generally useful and what is reported is reported well. However there are some important aspects to this type of technology that the report, as presented, does not address, and which would make the information much more useful to an international audience.
Firstly there is a whole section in the data analysis section that appears to be copied from instructions to authors, and requires deletion: The Materials and Methods should be described with sufficient details to allow others to replicate and build on the published results. Please note that the publication of your manuscript implicates that you must make all materials, data, computer code, and protocols associated with the publication available to readers. Please disclose at the submission stage any restrictions on the availability of materials or information. New methods and protocols should be described in detail while well-established methods can be briefly described and appropriately cited. Research manuscripts reporting large datasets that are deposited in a publicly available database should specify where the data have been deposited and provide the relevant accession numbers. If the accession numbers have not yet been obtained at the time of submission, please state that they will be provided during review. They must be provided prior to publication.
The assessment of accuracy of compounding the product appears reliable.
The two major aspects that need to be addressed, either with what the papers reviewed, stated about them, or reporting that the papers reviewed did not address them either. These are:
Acquisition and ongoing running costs; how do these compare with employing humans? Even if accuracy is improved and errors reduced, how many staff need to be replaced, doses prepared per year, to offset the acquisition and running costs of the devices?
Also, typically such equipment has a fixed set-up and strip-down/clean down time burden. As is seen in many other areas of compounding technology, these devices take a discreet, but largely fixed amount of time, and fixed cost of disposable equipment, to set the device ready for use plus another fixed time commitment to dismantle and clean down the equipment at the end of a session. This can exert a marked effect on the work throughput required for the machine to be time and cost effective, and to meaningfully interpret the time taken to complete a task. An example would be parenteral nutrition compounding devices; much quicker than manual methods to fill one PN prescription, but the time and cost of setting up and stripping down the devices make their use ineffective for any run of less than 10 filling operations. Even if any of these devices prepare the dose more quickly, after set-up and strip-down are accounted for, how many doses need to be prepared before there is an overall time saving?
During strip-down and cleaning, what are the risks of the operators being contaminated with product? Reducing operator exposure during compounding is of no benefit if they are exposed to contamination when cleaning and maintaining the device.
With these matters addressed or the need for them to be included in future studies discussed, I would welcome publication.
Author Response
Thanks for review to our manuscript and we uploaded response file with Word file. All authors appreciate the efforts and comments from the reviewers. We have endeavoured to respond to all the comments and to improve the readability and integrity of the manuscript.

Reviewer 2 Report
Dear authors,
thank you for submission of the manuscript.
I could not recognize additional information compared to the review of Batson et al.
Here some major issues regarding the submission:
From my perspective the Pico-SD and the RoBANS tool are not appropriate to analyse the published studies because these are not patient studies but experimental studies. Blinding is impossible. If manual preparation is analysed of course the unavoidable Hawthorn effect plays a role. The robot is always blinded. Thereby the methods used are not appropriate.
The limitations part is restricted to the outcome parameter cytotoxic contamination. And even according to this parameter the major limitation of a sensitive and accurate analytical method is not adressed. Neither the issue of official threshold limits is adressed. The limitations of the analytical methods used to determine accuracy of the preparation is not mentioned.
In the conclusion part the issue of cytotoxic contamination is not mentioned. Moreover, the conclusions in total are not matching the results and discussion part
Minor reasons:
Throughout the review terms dispensing robot, preparation robot and compounding robot are intercangeably used. In the introduction part mainly dispensing robot is used, what is definitely the wrong term.
ApothecaChemo is wrong, ApotecaChemo correct
Author Response

(The authors gave the same response as above.)

Reviewer 3 Report
Thank you for allowing me to review your manuscript. In the scope of meta-analyses, this is a fairly straight forward manuscript and analysis where there is significant heterogeneity in the study design, robots utilized, and data analysis performed … that being said, the authors clearly take this into appropriate consideration (e.g., restriction of contamination analysis to APOTECAchemo robot) and do not make any overreaching statements in their overall discussion and conclusions.
General comments:
** An additional publication of interest (dealing more in need for reducing workplace contamination) that would potentially add benefit to this manuscript: Zamboni WC, Salch SA, Cox J, Eckel S. It takes a village to raise awareness of and to address surface contamination of hazardous drugs. Journal of Oncology Pharmacy Practice. 2017;23(7):558-560. doi:10.1177/1078155217724650
** I believe that a general paragraph at the end of the Discussion (or Conclusion) stating what further study(ies) and/or endpoints that need to be evaluated would lend benefit to summarize the needs looking into the future for this pivotal concern.
Specific Comments:
** Lines 49 to 60: I think you can expand this to note how this isn’t even just a patient issue, but a pharmacist/hospital/OSHA workflow issue dealing with safety. Considering your second conclusion in the Discussion concerns workplace safety, I think it should at least be mentioned more prominently in the Introduction.
** Line 130 (Data Analysis): I think you need to include the various definitions of ‘contamination’, especially as you discuss multiple types without differentiating: workplace, microbial, drug.
** Lines 140 to 154: I am fairly sure this is reminder text on what needs to be included in the Meterials and methods and any datasets that would be required to upload. Remove from the manuscript unless it needs to be drafted for additional Data Analysis material.
** Line 157 (“searched”): I think you mean “initially identified”?
** Lines 306 to 324: It is actually surprising little info reported on the accuracy of these robots in real life settings given the hype since 2013 and the new USP requirements. Should comment on lack of data and spread of available data.
Author Response

(The authors gave the same response as above.)
